# Past conservation efforts reveal which actions lead to positive outcomes for species

Ashley T. Simkins [1]*, William J. Sutherland [1], Lynn V. Dicks [1], Craig Hilton-Taylor [2], Molly K. Grace [3], Stuart H. M. Butchart [1,4], Rebecca A. Senior [5], Silviu O. Petrovan [1]

1 Department of Zoology, University of Cambridge, Cambridge, United Kingdom, 2 International Union for Conservation of Nature Red List Unit, Cambridge, United Kingdom, 3 Department of Biology, University of Oxford, Oxford, United Kingdom, 4 BirdLife International, Cambridge, United Kingdom, 5 Conservation Ecology Group, Department of Biosciences, Durham University, Durham, United Kingdom

* ats43@cam.ac.uk

## Abstract

Understanding the consequences of past conservation efforts is essential to inform the means of maintaining and restoring species. Data from the IUCN Red List for 67,217 animal species were reviewed and analyzed to determine (i) which conservation actions have been implemented for different species, (ii) which types of species have improved in status and (iii) which actions are likely to have driven the improvements. At least 51.8% (34,847) of assessed species have actions reported, mostly comprising protected areas (82.7%). Proportionately more actions were reported for tetrapods and warm-water reef-building corals, and fewer for fish, dragonflies and damselflies and crustaceans. Species at greater risk of extinction have a wider range of species-targeted actions reported compared with less threatened species, reflecting differences in documentation and conservation efforts. Six times more species have deteriorated than improved in status, as reflected in their IUCN Red List category. Almost all species that improved have conservation actions in place, and typically were previously at high risk of extinction, have smaller ranges and were less likely to be documented as threatened by hunting and habitat loss or degradation. Improvements in status were driven by a wide range of actions, especially reintroductions; for amphibians and birds, area management was also important. While conservation interventions have reduced the extinction risk of some of the most threatened species, in very few cases has full recovery been achieved. Scaling up the extent and intensity of conservation interventions, particularly landscape-scale actions that benefit broadly distributed species, is urgently needed to assist the recovery of biodiversity.

## Introduction

Humanity is inextricably dependent on the rest of biodiversity for the range of functions and ecosystem services it provides. However, we are facing a global biodiversity crisis, with 28% of 150,388 assessed species threatened with extinction [1], and an estimated 1 million species facing this fate owing to human activities [2,3]. In December 2022, the Parties to the Convention on Biological Diversity (CBD) adopted a new Global Biodiversity Framework, the mission of which is "To take urgent action to halt and reverse biodiversity loss to put nature on the path

**Data availability statement:** All input data is available upon request from the IUCN Red List of Threatened Species for non-commerical purposes from https://www.iucnredlist.org/en. Code for analyses and proceeded data underpinning each figure is available at https://www.iucnredlist.org/resources/data-repository#Past%20conservation%20efforts

**Funding:** ATS is supported through the Natural Environment Research Council's C-CLEAR Doctoral Training Partnership (grant NE/S007164/1). The funders had no role in study design, data collection and analysis, decision to publish, or preparation of the manuscript.

**Competing interests:** LVD is a Board Member of Natural England. All other authors have declared that no competing interests exist.

**Abbreviations:** CBD, Convention on Biological Diversity; CR, Critically Endangered; DD, Data Deficient; EN, Endangered; EW, Extinct in the Wild; EX, Extinct; GSS, Green Status of Species; LC, Least Concern; NT, Near Threatened; VU, Vulnerable.

to recovery…" [4]. This includes commitments to halt human-driven extinctions, reduce the extinction rate and extinction risk of all species 10-fold, and increase the abundance of wild species to healthy and resilient levels. Urgent conservation actions are necessary to achieve these outcomes [4,5].

Crucially, we need effective conservation based on evidence [6], so that actions can be transparently assessed and implemented in different contexts. There is ample evidence that conservation action can work [7–10] and that targeted efforts for species are needed [11]. However, this evidence is often presented in disparate sources that can be hard to access [12]. There are also many gaps in the evidence base, including biases towards particular species (e.g., birds and mammals), geographies (e.g., North America and Europe) [13,14] and actions (e.g., protected areas) [15]. Conservationists often lack sufficient time, capacity and/or incentives to publish outcomes of conservation actions in the scientific literature; there is a need to consolidate practitioners' knowledge to understand which actions have been linked to improvements in species status. This is critical to inform the targeting of future conservation actions and for tracking progress towards halting global biodiversity loss.

The IUCN Red List of Threatened Species (hereafter Red List) is the most comprehensive source of information on species conservation [16], including actions needed and underway. It incorporates input and varied information from a wide range of stakeholders including conservation practitioners [17] and is useful for assessing conservation impact, including written accounts (narrative fields) and tabular data relating to each species' status and the actions that are in place to promote their conservation [18]. Correlative analyses have used these data, for example to explore the relationship between population trends or changes in species' status and different conservation actions [19,20]. However, although the tabular data on conservation actions can be easily analyzed, some information in the narrative fields may not be reflected in the tabular fields [21,22], posing challenges for multi-species analyses, particularly those that extend beyond particular subsets of species [23,24].

When reassessment of species on the Red List leads to a revision in their Red List category, the reasons for this are documented in order to distinguish 'genuine' changes in category, (e.g., resulting from increases or decreases in population size, rate of decline or distribution) from 'non-genuine' change (e.g., resulting from revisions to taxonomy or improvements in knowledge of population or range size and trends), and often includes the driver of this change. These genuine changes underpin the Red List Index [25–27]. Associated documentation on drivers of genuine changes enable exploration of the relationship between threats and actions on species outcomes. For a small but growing number of species, Red List assessments also include documentation of recovery and conservation impact using the Green Status of Species approach [28,29]. Given the breadth of species assessed and the inclusion of practitioner knowledge, the Red List can help us gain a more comprehensive understanding of what has worked in species conservation.

We leveraged these data from the Red List to explore the following three questions: (i) which conservation actions have been implemented for different species, (ii) which species have improved in conservation status and (iii) which actions were associated with these improvements?

## Methods

### Data sources and handling

Information for all assessed animal species was downloaded from the Red List version 2023.1 [1], including their taxonomic classification, Red List category of extinction risk, ecosystem type (terrestrial, freshwater and/or marine), threats, conservation actions in place, global

population trend and generation length (the average age of breeding individuals, in years), along with their range maps. Extinct species, taxa belonging to groups that were not comprehensively assessed (i.e., for which less than 80% of the species have Red List assessments), and subspecies and subpopulations were excluded from analyses. The resulting data included 7,983 amphibians, 11,038 birds, 1,240 cartilaginous fish, 750 cephalopods, 6,223 dragonflies and damselflies, 80 hagfish, 4 horseshoe crabs, 38 lampreys, 8 lobe-finned fish, 5,895 mammals, 10,222 reptiles, 14,347 freshwater fish, 2,886 selected crustacea, 687 selected gastropods, 4,988 selected marine fish and 828 warm-water reef-building corals (see S1 Text for classification of selected groups).

Species' range size (in km$^2$) was calculated by summing the total area of their range coded as Extant, Probably Extant or Possibly Extinct (presence), Native, Reintroduced or Assisted Colonization (origin) and Resident or Breeding (seasonality) codes, in cylindrical equal area projection. For migratory species with breeding ranges, their range size reflected their breeding (or where applicable breeding and resident) range; non-breeding ranges were excluded to avoid over-inflation of range size estimates. The area of species' mapped range was used rather than population size, as it was reported for a much greater proportion of species, with range having been shown to be correlated with (and often used as a proxy for) population size [17]. Generation lengths were quantified in a range of formats; where available, the best estimate was used, while mean values were used if a range was given, with non-numeric values reviewed and converted to numerals accordingly. Just under half of mammals did not contain information about their generation length, so this was supplemented using data from Pacifici and colleagues [30].

Threat and action types reported in tabular form were aggregated respectively to simplify interpretation of analyses by combining similar concepts (see S2 and S3 Texts, respectively, for details). Future threats and those coded by Red List assessors as having negligible impact were excluded. This left the following threats: habitat loss and degradation, hunting or fishing, problematic or invasive species or diseases, pollution and climate change; and actions: potential occurrence in protected areas, area management plan, control of invasive or problematic species or diseases, species management, reintroduction or translocation, awareness and education, international legislation or trade control, captive breeding or monitoring. Whether sites of importance (e.g., Key Biodiversity Areas) have been identified for each species was excluded due to expert assessors' information suggesting it has been applied inconsistently between groups.

The genuine changes in species' Red List category we analyzed were based on two sources. The first was data that underpin the Red List Index [1,27], which includes the category change, timing and a text narrative justifying the reason for improvement or deterioration in status (typically threats or actions). This was combined with information on additional genuine changes in the period 2016–2022 recorded by the Red List Unit but that had not yet been published in RLI assessments. For genuine improvements in status, documentation of the reason for the improvement was reviewed against the coded actions in place for each species. If the specific action was not recorded (e.g., some simply referred to "conservation action"), the species' Red List assessment and "actions in place" text accounts were reviewed to see if these attributed particular actions to improvements in species status. Where this was absent, and for cases where the action described did not fall within the options for actions in place (e.g., livelihoods, economic or other incentives), they were classified as "other action". When there was no clear indication that the improvement in category was driven by conservation action, or where it was attributed to natural processes (e.g., amelioration of drought, habitat succession following land abandonment, etc.), this was recorded as such.

Information from Green Status of Species assessments (a measure of species recovery) was extracted for the 35 animal species on the Red List with published assessments as of March 2024 (including *Trechus terrabravensis* and *Troides helena*, from groups that are not comprehensively assessed). These include estimates of the impact of conservation compared with a counterfactual of no action. This counterfactual is also spatially explicit, examining the impacts of conservation in distinct parts of the species range (spatial units). Within each spatial unit, the species' current state (Absent, Present, Viable, or Functional) [29] is estimated, as well as the counterfactual state (i.e., the expected current state without conservation action). For each species' spatial unit, the best estimate for both the current and counterfactual state were extracted, along with documented past and current threats, and past and current conservation actions. Note this also included the action of livelihood, economic and other incentives (as this is a category in the Conservation Actions Needed classification scheme but not the Actions in Place) [18].

## Analysis of conservation actions in place

For each species group, the number of species with (i) no reported conservation action, (ii) at least one conservation action and (iii) each coded conservation action, was calculated. Chi-squared analysis and post hoc tests were used to determine whether the proportion of species with or without any conservation action(s), and for each conservation action type in turn, varied according to Red List category and taxonomic group, and if so, how. Species assessed as Data Deficient (DD) and Extinct in the Wild (EW) were excluded because DD species lack sufficient information to assess their extinction risk, and EW species typically have captive breeding but few if any other actions in place. Species assessed as Least Concern (LC) were excluded from the chi-squared analysis across Red List categories as it is optional to report actions for LC species, so it is not possible to distinguish a genuine lack of actions from lack of documentation.

## Models of improvements in species' conservation status

Three different indicators were used to investigate which species' traits, threats and actions are associated with improvements in species' conservation status, focusing on risk of extinction in the wild. (1) Species global population trend: declining, stable or increasing (excluding those with unknown trends), estimated over a 5-year window around the date of assessment (for all comprehensively assessed species). (2) Net (overall) genuine change in Red List category: species were coded as "uplisted", "unchanged" or "downlisted" based on whether their most recent assessment category was a deterioration, unchanged, or an improvement compared with their first assessment category (for amphibians, birds and mammals). As before, LC species were excluded due to inconsistencies in reporting for these species. Currently EW species were excluded from the population trend model as they have no population trend in the wild until they are reintroduced (the Red List category model uses their former category so EW species can be included). (3) Prevented declines in species' state within specific spatial units, coded as a 'prevented decline' and otherwise coded as 'no impact' based on the changes between the documented current and the counterfactual state (without conservation action) for each spatial unit (species' spatial units that were classified as "functional" in both current and counterfactual states in the analysis) were excluded as they could not improve. The population trend model included all comprehensively assessed animal groups, the Red List category change model included amphibians, birds and mammals, and the prevented decline in species state model included the 35 animal species with published Green Status of Species assessments. These three indicators were modeled against a range of species traits, threats and actions (see Table 1).

Species' range size (log km$^2$) was included as it is a key indicator of extinction risk [17,31], logged given most species have small range sizes (so to normalize the distribution of the data).

**Table 1. Structure of three models of improvement in conservation status against a range of variables related to species traits, threats and conservation actions. The five threats consist of: habitat loss/degradation, hunting/fishing, invasive/problematic species/diseases, pollution or climate change. The seven actions consist of: potential occurrence in protected areas, area management plans, invasive/problematic species control, species management plans, reintroduction/translocation, international legislation/trade control, education/awareness raising.**

| Response variable | Model type | Fixed effects | Random effects | Taxonomic groups | Excluded cases |
|---|---|---|---|---|---|
| (1) Global population trend<br>*levels:*<br>*Increasing>*<br>*Stable>*<br>*Declining* | Cumulative linked mixed model | log(range) + current IUCN Red List category + system + ecosystem + 5 threat types + 7 action types | Taxonomic group | All comprehensively assessed taxa | Least Concern species, Extinct in the Wild species and species with unknown population trends |
|  | Cumulative linked model | As above, plus log generation length for birds and mammals | N/a | Amphibians, birds, freshwater fish, mammals and reptiles separately | As above plus ecosystem excluded for except mammals, Freshwater fish also excluded legislation as only reported for declining species |
| (2) Genuine change in IUCN Red List category<br>*levels:*<br>*Downlisted>*<br>*Stable>*<br>*Uplisted* | Cumulative linked mixed model | log(range) + initial IUCN Red List category + ecosystem + 5 threat types + 7 action types | Taxonomic group | Amphibians, birds and mammals | N/a |
|  | Cumulative linked model | As above, plus log generation length for birds and mammals | N/a | Above groups separately | Extinct in the Wild species for birds as too few reported across groups |
| (3) Prevented declines (based on Green Status of Species)<br>*levels:*<br>*Prevented declines>*<br>*No change in state* | Generalized linear mixed model (logistic) | log(generation length) + log(range) + counterfactual state + ecosystem + 5 threat types + 7 action types + livelihood/economic incentives | Species name | 35 species with published GSS assessments | N/a |

Species' ecosystem type (terrestrial, marine or both terrestrial and marine) was included to account for differences in threats and actions between the different realms. Freshwater was treated as part of terrestrial in order to focus on the distinction between land/freshwater and sea, between which the different species' traits, threats and actions are likely to differ to a higher degree. The other factors included in the model were the presence/absence of the five threats (habitat loss or degradation, hunting or fishing, invasive or problematic species or diseases, climate change and pollution) and seven actions (potential occurrence in protected areas, area management plans, control of invasive or problematic species or diseases, species' management, reintroduction or translocation, awareness and education, and international legislation or trade control). Monitoring and captive breeding were excluded as the former is not a conservation intervention per se, and the latter does not impact species' extinction risk in the wild until individuals are introduced (which is documented separately), though of course play key roles in preventing overall species extinction and facilitating reintroduction to the wild.

Species' current Red List category (for the population trend model) or initial Red List category (for the genuine Red List category changes model) was also included as an ordinal factor, where LC < Near Threatened < Vulnerable < Endangered < Critically Endangered < EW. The prevented declines in state model included the counterfactual state within each spatial unit, and an additional action of livelihood, economic and other incentives. Taxonomic group was included as a random factor in the models for population trend and genuine Red List category changes. Due to the limited number of species in the Green Status of Species model, species name was included as a random factor rather than taxonomic group because each species is assessed in multiple spatial units, leading to multiple observations for each species in the model.

The population trend and genuine category change models were also run independently for each taxonomic group with sufficient data, and for species traits. This also included LC

species as this information should be reported for LC species similarly to other threat levels. Logged generation length (another important indicator of extinction risk [17], logged given most species have short generation times) was only included for birds, mammals, trait-only and prevented declines in species' state models due to limited data availability.

## Modeling

Cumulative linked mixed models were used to model the population trend and genuine change in extinction risk, and generalized linear mixed models with a logistic function were used to model prevented declines in species' state in spatial units (as only had two categories rather than three). Backward selection was used to find the best model, dropping the variable with the highest $p$-value >0.05 in turn until all remaining variables were significant [32]. Both backwards and forwards Akaike information criterion, corrected for small sample size selection were also undertaken for the three global models as a model selection sensitivity analysis (S4 Table) [33]. Model estimates were then visualized in a heatmap to enable visual comparison of the relationships of different variables with the different indicators of improvement in status within and between the various models.

To explore how many species have undergone net changes between each Red List category, the numbers of species moving from and to each category was also visualized in a contingency table. To visualize the distribution of species that underwent genuine changes in Red List category, their ranges were loaded into QGIS Desktop version 3.22.12 [34]. Ranges of species that underwent genuine net improvement or deterioration in Red List category were then in turn spatially joined to a fishnet grid (at 2 decimal degree resolution), to count the number of species per grid cell that improved or worsened in category. Country boundaries were from the Database of Global Administrative Areas [35].

## Actions driving genuine changes in Red List category

For species that underwent genuine changes in Red List category, we calculated the number of species with (i) no action in place, (ii) at least one action in place, (iii) each of the possible actions in place (excluding the 'other conservation actions') and (iv) each action in place that was judged by assessors as driving the genuine change in Red List category. This was done for birds and mammals independently, as the only groups with data on genuine changes in Red List category with assessor judged reasons for genuine improvements. Chi-squared and post hoc tests were used to evaluate whether there were differences in the types of actions that led to improvements relative to those in place for each taxa and independently.

## Software

All analysis was undertaken in RStudio [36] using R version 4.3.1 [37]. This included use of the following packages for data manipulation: tidyverse [38]; statistical analysis: chisq.posthoc. test [39], lme4 [40], ordinal [41], ggcorrplot [42], MuMin [43]; plotting: ggplot2 [44], pheatmap [45]; and spatial analysis: sf [46,47].

## Results

### Numbers of conservation actions in place for species

Over half of species (51.8% or 34,848 of 67,217) in comprehensively assessed animal groups had documented conservation actions in place, rising to 58.7% of (12,574) threatened species. Tetrapods, warm-water reef-building corals and Near Threatened species were more likely to have at least one conservation action reported compared with invertebrates, aquatic species (Fig 1A; S1 Table; $X^2$ = 15,362, df = 15, $p$ < 0.001) or Critically Endangered species (S1 Table;

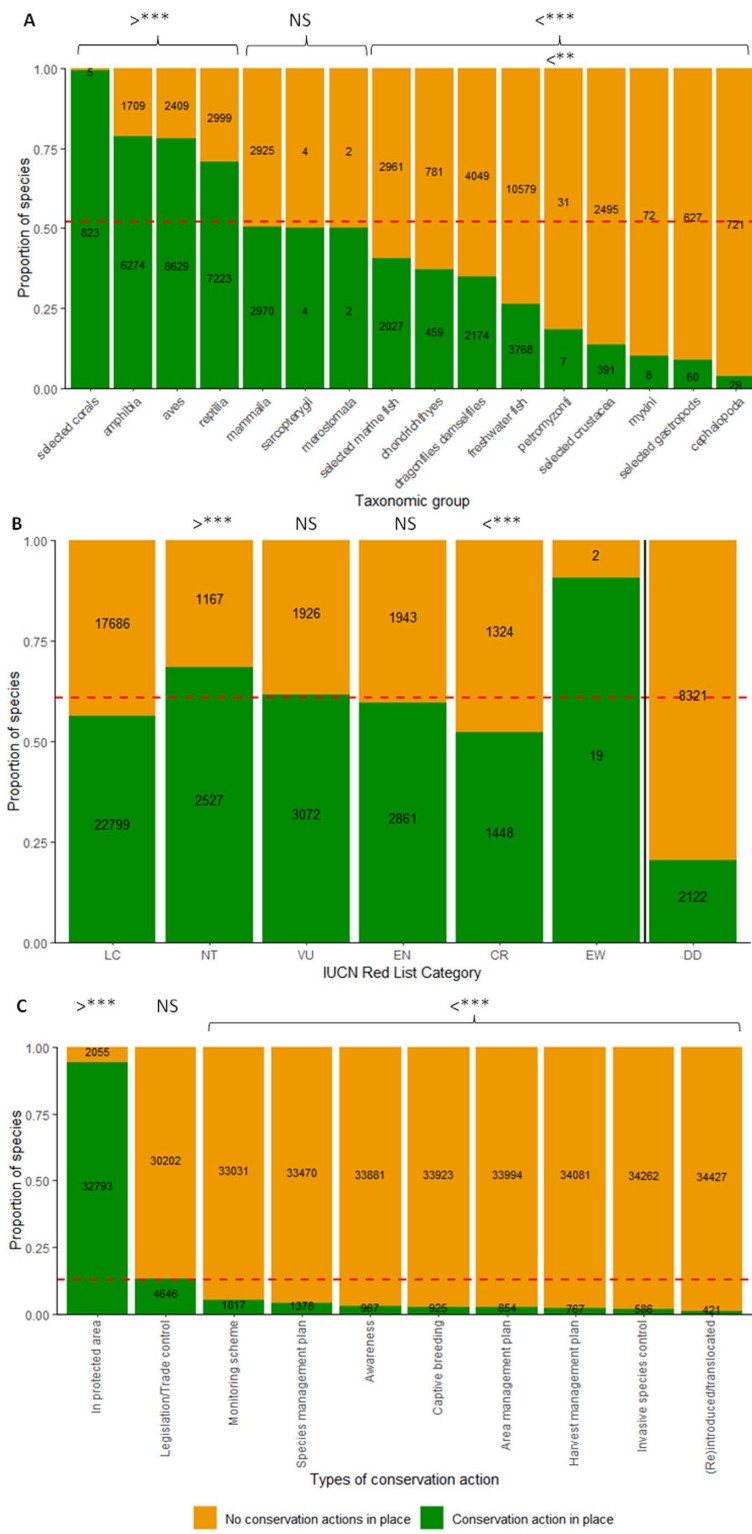

**Fig 1. The proportion of species (in comprehensively assessed taxonomic groups) with at least one conservation action documented as being in place, for (A) different taxonomic groups, (B) different IUCN Red list categories of extinction risk, and (C) for different conservation actions.** The dashed orange line indicates the mean proportion across all species (in B; LC, EW and DD were not evaluated). The > and < signs indicating significantly more or less

likely to be in place compared to other actions (evaluated using a chi-squared test in S1 Table), with statistical significance level indicated as follows: *$p < 0.05$, **$p < 0.01$, ***$p < 0.001$ and $p > 0.05$ NS. See Fig 1A.csv, Fig 1B.csv and Fig 1C.csv respectively for underlying data, available at: https://www.iucnredlist.org/resources/data-repository#Past%20conservation%20efforts.

$X^2 = 179.15$, df = 3, $p < 0.001$), with some variation across Red List categories in species' groups (S1 Fig). Freshwater fish (41.0%) and amphibians and reptiles (26.3%), and species in the Neotropics (29.3%), Indomalaya (21.1%) and Afrotropics (18.4%) make up the majority of the 5,193 threatened species without documented actions (1,324 of which are Critically Endangered).

## Types of conservation actions in place for species

Potential occurrence in protected areas was the most common action reported (94.1% of 34,848 species, Figs 1C and S2A; S2 Table; $X^2 = 229,474$, df = 9, $p < 0.001$). Birds and mammals were also more likely to have legislation or trade control, and cartilaginous fish and crustaceans were more likely to have harvest management in place compared with other action types respectively. Tetrapods, particularly birds and mammals, and species at greater risk of extinction typically had a wider variety of actions documented, particularly species-targeted actions, compared with aquatic and invertebrate groups (S2A Fig) and lower risk species (S2 Table; S2B Fig; $X^2 = 516.79$, df = 27, $p < 0.001$), which mostly have only potential occurrence in protected areas documented.

## Species with improvements in conservation status

Nearly six times as many amphibians, birds and mammals underwent net deterioration (1,220 species) in Red List category than net improvement (201 species), with most species moving one category (Table 2). Twenty-five species deteriorated from LC to Critically Endangered, with zero species improving to the opposite extent. Amphibians, birds and mammals that underwent genuine deterioration in Red List category were found across the tropics, southern Europe, parts of central Asia and south-eastern Australia, with highest concentrations in the tropical Andes, Peninsula Malaysia, Sumatra and Borneo (Fig 2A).

Most species with globally increasing populations (78.3% of 969 species) or that improved in Red List category since 1980 (99.3% of 288 species) have a conservation action in place. Species that underwent genuine improvements in Red List category were 7.5 times less concentrated than those that deteriorated, with the highest numbers of species on islands (e.g., New Zealand, Mauritius, the Seychelles, Chatham Island, Guadeloupe and Borneo), as well as

**Table 2. The total number of species which started and ended in each respective combination of IUCN Red List categories between 1980 and 2024. Net genuine improvements in category are written in green, deteriorations in orange, and unchanged in black.**

| | | Previous IUCN Red List Category | | | | | |
|---|---|---|---|---|---|---|---|
| | | LC | NT | VU | EN | CR | EW |
| Current IUCN Red List Category | LC | 15,711 | 25 | 8 | 1 | 0 | 0 |
| | NT | 336 | 1,416 | 46 | 30 | 10 | 0 |
| | VU | 144 | 149 | 1,738 | 29 | 15 | 0 |
| | EN | 45 | 50 | 146 | 1914 | 34 | 1 |
| | CR | 25 | 21 | 92 | 207 | 925 | 2 |
| | EW | 0 | 0 | 0 | 0 | 5 | 3 |

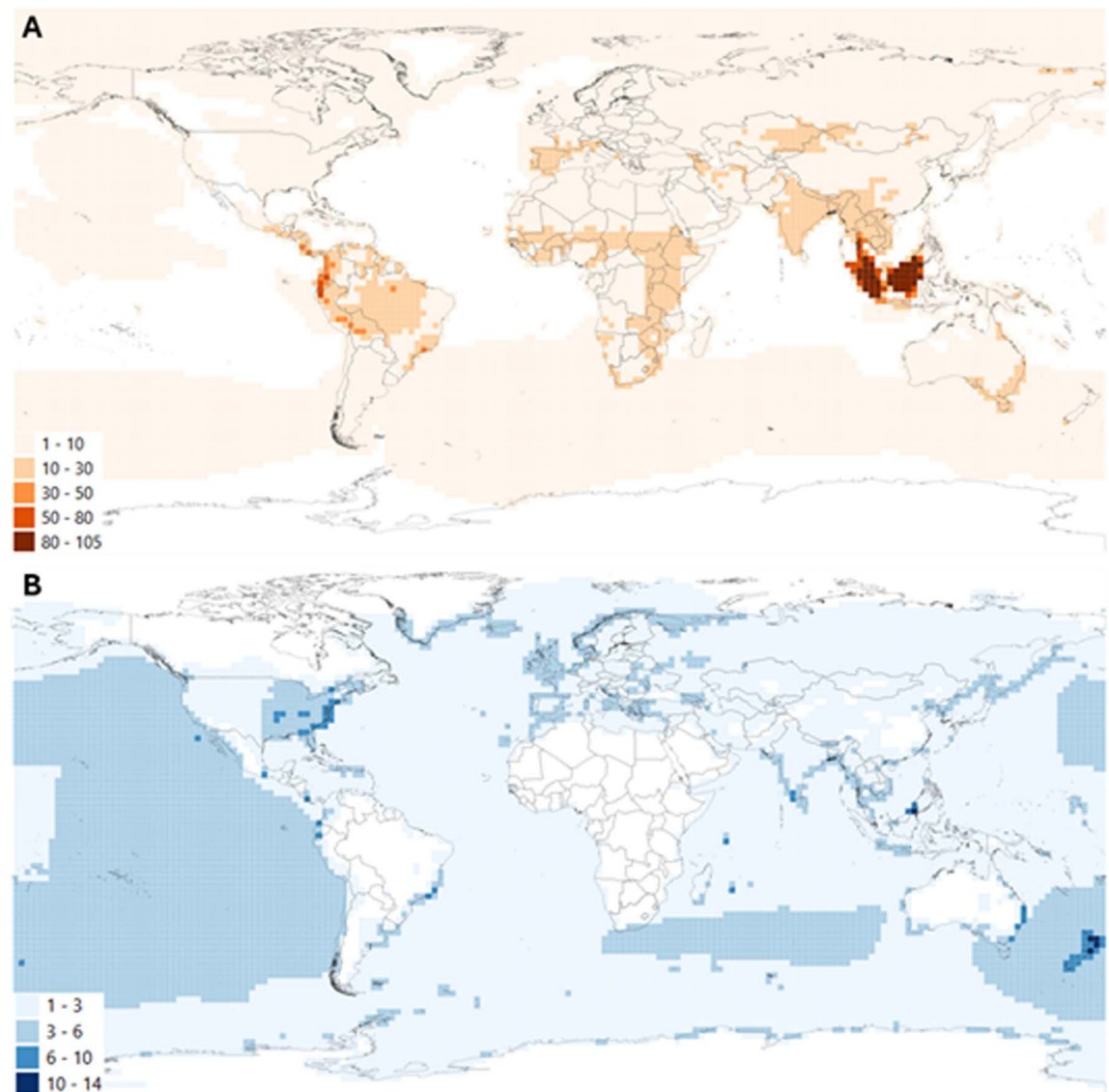

**Fig 2. Species richness map of species that have undergone genuine deteriorations (orange; A) or genuine improvements (blue; B) in IUCN Red List category (measure of extinction risk).** Lighter colors indicate fewer species and darker colors indicator high numbers of species. White indicates areas where no species underwent a change in IUCN Red List category. See Fig 2A.shp and Fig 2B.shp respectively for underlying data, available at: https://www.iucnredlist.org/resources/data-repository#Past%20conservation%20efforts. The country boundaries are from GADM [35], available for academic and non-commercial purposes here: https://gadm.org/license.html.

in parts of the eastern United States of America, Costa Rica, eastern Australia, the southern tip of India and particular locations in Mexico, Ecuador and Brazil's Atlantic Forest (Fig 2B). It should be noted that values in coastal regions may be slightly inflated owing to overlaps with marine distributions of several seabird and whale species' ranges.

## Species' traits

Species with smaller global ranges, those at lower risk of extinction and those found in both terrestrial and marine ecosystems were more likely to have increasing global populations.

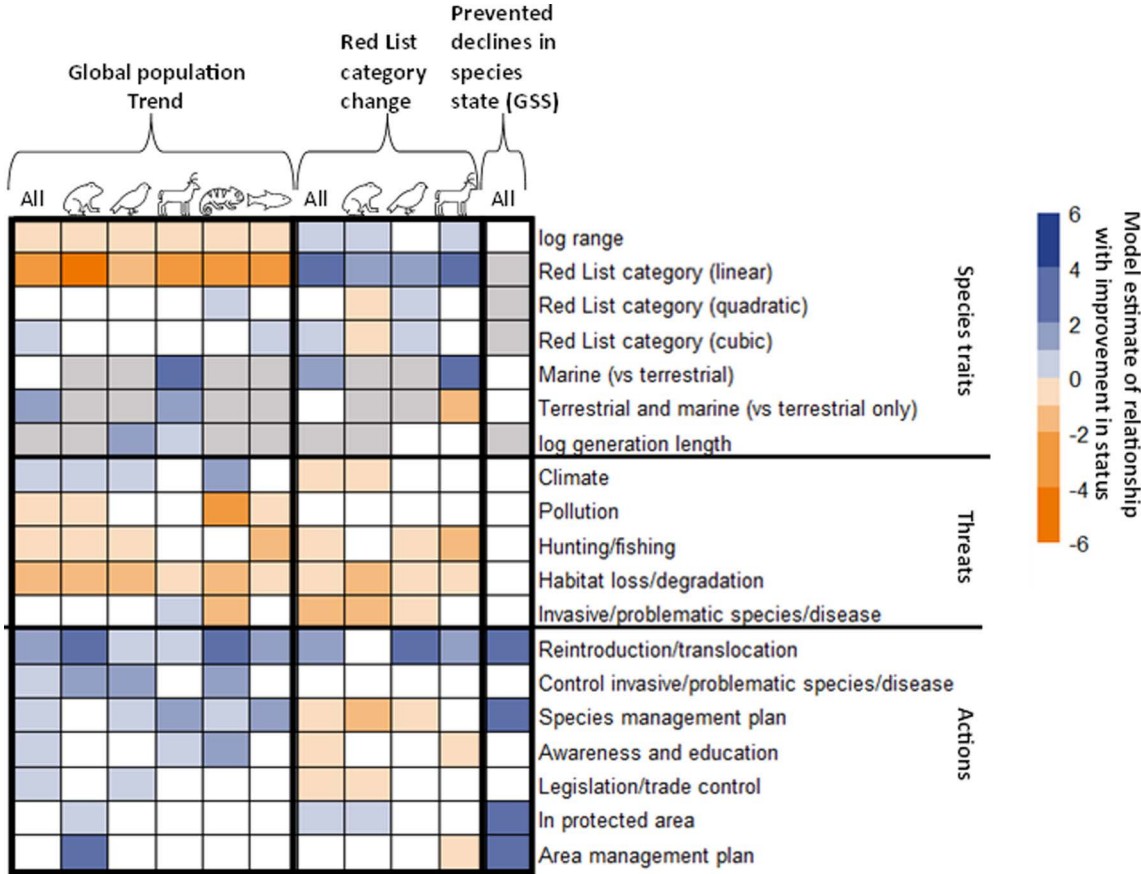

**Fig 3. Estimated relationship between various species traits, threats and actions in place with three indicators of improvement in species status; species' global population trends, genuine changes in IUCN Red List category (both derived from cumulative-linked mixed models) and prevented declines in species' state in spatial units (from the Green Status of Species; derived from a Generalized Linear Model).** Modeled outputs are shown across all species and for different groups of species in turn (amphibians, birds, mammals, reptiles and freshwater fish). Gray indicates variables that were not included in the initial model, and white indicates variables that were dropped through backwards selection (though if variables were not included or dropped for all, these were not shown in the plot). Darker colors indicate largest variable estimates, with blue indicating larger positive associations and orange larger negative associations with improvements in species status. Note the population trend model included current Red List category, and the genuine Red List category change model included initial Red List category (before change). See S3 Table for full model outputs.

Species with larger global ranges, those previously at higher risk of extinction and marine species were more likely to have improved in Red List category. Generally, species with shorter generation times were more likely to have increasing global population trends and have improved in Red List category, except for birds and mammals, for which species with longer generation times are more likely to have increasing global populations (Fig 3; S3 Table).

## Threats

Species threatened by habitat loss or degradation and hunting or fishing were more likely to be undergoing population declines or worsening in Red List category. Species threatened by pollution were more likely to have declining populations, and those threatened by invasive or problematic species, disease, or climate change were more likely to have deteriorated in Red List category. By contrast, species threatened by climate change and mammalian species threatened by problematic or invasive species or diseases were more likely to have increasing populations.

## Actions

Species that have been reintroduced or translocated were more likely to have improved in status across all three indicators. Species with species management plans were more likely to have increasing populations and have experienced prevented declines in state, and those potentially occurring in protected areas were more likely to have improved in Red List category and have experienced prevented decline in state. Species with invasive or problematic species or disease control, education or awareness raising or international legislation or trade control, and amphibians potentially occurring in protected areas and those with area management plans, were more likely to have increasing populations. Species with area management plans were more likely to have had experienced prevented declines in state. Contrastingly, species with species management plans, awareness or education raising or legislation or trade control, and mammals with area management plans were more likely to have deteriorated in Red List category (Fig 3; S3 Table).

## Drivers of genuine improvements in Red List category

Conservation actions were attributed to 71.1% (91) of mammal and bird species Red List category improvements; 63.7% (58) of which had ≥2 conservation actions attributed. Only 5.5% (7) of improvements were attributed to non-conservation reasons, with the remainder having no clear attribution (S4 Text).

Potential occurrence in protected areas and species management plans were the most reported action for birds that improvements in Red List category. Area-based regional management plans, reintroduction or translocation and invasive or problematic species or disease control were the most attributed actions, with legislation and trade control and monitoring schemes less likely to be attributed, relative to how often they were in place ($X^2$ = 75.418, df = 9, $p$ < 0.001; Fig 4A; S5 Table). Potential occurrence in protected areas was the most common action reported for mammals that improved in Red List category, with reintroductions or translocations and harvest management the most proportionately attributed, although no significant difference in attribution was detected ($X^2$ = 9.509, df = 7, $p$ = 0.218; Fig 4B).

## Discussion

### Which conservation actions have been implemented for different species?

More than half of all species and at least 59% of threatened species in comprehensively assessed animal groups have documented conservation actions in place. More species likely have actions underway, as reporting in place actions is recommended (not required) in the Red List, and is optional for LC species. Actions are also often mentioned in the Conservation Action narrative but not reflected in the tabular data [21,22]. Higher proportions of tetrapods, particularly birds, with reported actions likely reflects taxonomic biases in documentation, data availability and implementation [48]. For example, all birds have been assessed eight times by BirdLife International, whereas mammals have only been assessed twice by more than 35 different groups, and amphibians and warm-water reef-building corals the only other animal groups to have been completely reassessed.

Most reported actions are potential occurrence in protected areas, often the sole action for species including those in more neglected groups or at lower risk of extinction. This likely reflects rapid ease of assessment (requiring only a spatial analysis of species' distribution maps and protected area boundaries, which are readily available), and so is likely disproportionately represented. Protected areas and international legislation can also easily be assessed using global databases [49,50], likely explaining the prevalence of these actions in the dataset.

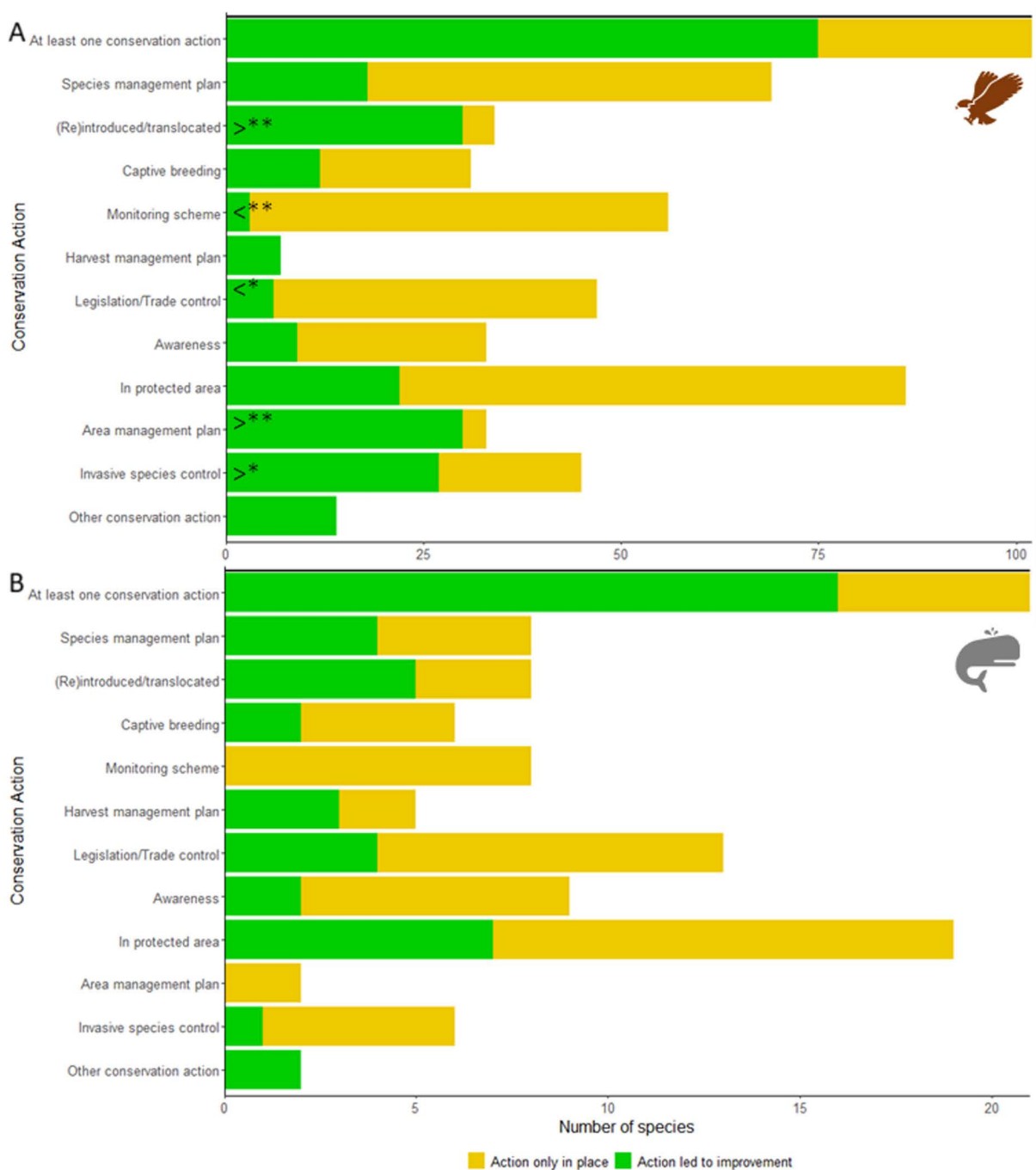

**Fig 4. Actions in place for (A) birds and (B) mammals that have shown a genuine improvement in IUCN Red List category (and that have at least one conservation action in place), and whether any of those actions were deemed by experts to have contributed towards the genuine improvement.** Asterisks indicate where actions were responsible for improvements in IUCN Red List category more than expected (determined by chi-squared analysis; S5 Table). See Fig 4.csv for underlying data, available at: https://www.iucnredlist.org/resources/data-repository#Past%20conservation%20efforts.

Proportionately more Critically Endangered species have species-targeted actions reported, possibly reflecting greater targeting and or documenting of conservation efforts for species most urgently requiring them [20,22].

Despite this, our analysis probably exaggerates the degree to which conservation actions for species have been targeted and implemented throughout their ranges. While some protected areas were designated and/or are managed to conserve particular species, most are not and instead aim to protect a particular location and its ecological community [51,52]. Protected areas are biased to less accessible locations under lower human-pressure and do not adequately safeguard many species from threats [53,54], or may lack effective management [55]. Furthermore, given the large coverage of the world by protected areas (17.6% of terrestrial and island waters and 8.4% of marine areas) [56], it is likely many species' ranges overlap protected areas only marginally [57], with recent studies showing around 91% of threatened species have insufficient representation of their habitats within protected areas [22].

Given these issues, we emphasize the importance of documenting actions underway for all species when undertaking Red List assessments, as well as recording the spatial extent and impact of actions, to enable more effective prioritization of conservation actions. Critically, the 5,194 (41.3% of) threatened species with no reported actions need urgent evaluation to determine if there are genuinely no actions in place, and if so, they should be urgent priorities for conservation attention and action.

## Which species have improved in conservation status?

The number and spatial extent of the distribution of species that have deteriorated in status greatly exceeds that of species that are recovering. The former are concentrated in threatened species' hotspots [58], whereas species that have improved in status are generally distributed on islands or small habitat patches, perhaps because threats are more tractable to mitigate in restricted locations [19]. Some species have deteriorated from the lowest to highest extinction risk categories on the Red List, but conversely no species has fully recovered from near extinction, suggesting that we can prevent extinction but have yet to fully recover species formerly on the brink of extinction.

Species with smaller ranges and previously at higher risk of extinction are most likely to have improved in status, perhaps reflecting more circumscribed conservation challenges and greater targeting of conservation efforts towards species most at risk [20]. This may also relate to the Red List criteria, with rarer species having greater potential to increase population numbers above thresholds needed for downlisting compared with more abundant species. Similarly, as we considered current (not former) range, past range expansions and contractions may explain status improvement and deterioration in currently species with larger and smaller ranges respectively. It is worth noting that we assumed range size to be a suitable proxy for population size, but this relationship is imperfect, with variation across species, so the exact relationship between population size and Red List category may differ.

Species (particularly mammals) in marine ecosystems were more likely to have improved in status than those in terrestrial systems, perhaps due to less concentrated area-competition with people compared with on land or improvements in fishery and harvesting regulation. Generally, species with shorter generation times were more likely to have improved in status, as they tend to have higher reproductive potential and therefore recovery capacity. However, bird and mammal species showed the opposite, perhaps owing to greater conservation attention and resources to larger-bodied species [59,60] which typically have longer generations, such as the European bison (*Bison bonasus*); [61].

Unsurprisingly, species that have or are deteriorating in status were more likely to be threatened by one of the five major threats, particularly habitat loss/degradation and hunting/fishing which also threaten the greatest proportion of species [1]. The fact that some species with reported threats have or are improving in status may reflect variation in trends and threats across species' ranges, mitigation or offset of the impacts of the threats on species, or the inclusion of threats with unknown impacts (because the timing of their impact is uncertain).

## Which actions were associated with or attributed to improvements in species status

Almost all species that have or are improving in status have conservation action(s) documented, with most status improvements attributed to conservation, and all conservation actions attributed to improvement of one or more species. This reinforces others' findings that conservation works [10,62].

Unsurprisingly, we found species that have been reintroduced or translocated were consistently more likely to have or be improving in status, and when in place these actions were most often attributed to improvements. When these interventions are successful, they can significantly increase the wild population or distribution of a species and therefore reduce extinction risk [17,63]. For example, intensive captive breeding and reintroduction lead to the recovery of the Mauritius Kestrel (*Falco punctatus*), whose population increased from 4 to >250 individuals leading to downlisting from Critically Endangered to Vulnerable [64]. Species with species-targeted management plans in place were also more likely to have or be improving in status, reinforcing the need for species-targeted actions for many species [11]. Similarly, the association and attribution of the control and eradication of invasive alien species to improvements in species status is consistent with the many reported successful management programmes, particularly on islands [65] such as the recovery of Campbell Teal (*Anas nesiotis*) [66] following successful eradication of invasive rats from Campbell Island.

Particular actions may also benefit certain groups of species disproportionately. Amphibian species with site-based actions such as area management plans and protected areas were more likely to have or be improving in status, with area management often attributed to status improvement in bird species, reinforcing other's findings [20,67]. Amphibian species typically have smaller ranges compared with other terrestrial vertebrates, making it easier to target such interventions. In addition, effective management of important sites for species of conservation concern (e.g., Important Bird and Biodiversity Areas and other Key Biodiversity Areas), whether through formal protected areas, other effective area-based conservation measures, or other means, can generate substantial benefits for species for which site-scale conservation is appropriate [68,69].

However, in some cases, conservation actions may fail to positively impact species (which may explain non-significant or negative results), such as failed attempts to eradicate invasive alien mice on Gough Island [70] or rats (*Rattus* spp.) on Henderson Island [71], or protected areas that exist on paper only, and are not implemented effectively on the ground. The scale and duration of implementation of the action is important: in some cases actions may not have been carried out at sufficient scale, extent or effort to benefit the species sufficiently for any improvement to lead to a reclassification of Red List category, or there may be time-lags before species' recover sufficiently. Attempts to promote recovery may be insufficient if the threats impacting the species are not addressed sufficiently, such as the unsuccessful reintroduction of the Arabian oryx (*Oryx leucoryx*) in Oman due to hunting pressure [72]. In other cases, a threat may be effectively mitigated but there may be other threats or additional actions may be required; we found two or more actions were attributed to status improvements in species that improved Red List category. This includes the Western quoll (*Dasyurus geoffroii*) which improved in status following translocation, invasive predator (feral cats) control, and public awareness raising [73].

Discrepancies between the results of our models may arise from differences in the species and species' groups included, the indicator resolution, and the temporal and spatial scales of the data. This varies from all comprehensively assessed animal groups and species (in the population model) to 35 species (in the prevented declines model), global (in the population and Red List category models) to distinct species populations within their ranges (in the prevented declines model), and from the past five years (in the population model) to up to 1,500 years ago (in the prevented declines model).

## Considerations when using the Red List to understand impact

The Red List has been designed as a standardized, objective and transparent way to classify species' risk of extinction, rather than specifically for impact evaluation [17]. Our analysis demonstrates its utility for this, but it is not without limitations, such as those relating to taxonomic biases in assessed species, frequency of reassessment, and levels of documentation for each species. Both Red List and Green Status of Species (GSS) assessments also rely on expert assessment, with analyses derived from Red List data and GSS assessments typically not resulting from empirical tests of the outcomes of actions, but instead consider relevant post hoc and somewhat subjective assessments of counterfactual states [7,62,74–76].

Relatively few species have undergone documented genuine Red List category changes, underestimating the true number of species improving due to the categorical quantification of extinction risk in the Red List, time-lags between updates and documentation biases. As the Red List categories are based on quantitative thresholds, a species can improve or deteriorate in status (sometimes substantially) without changing category. Even when a species does recover sufficiently to cross thresholds to qualify for a category of lower extinction risk, it must qualify at this level for five years before being 'downlisted' in the Red List [1] (though this will be captured as a genuine change). The system is designed to enable species to be assessed even with limited data availability and to account for uncertainty [17], but as a result masks finer scale impacts of conservation efforts.

Even for the four animal groups in which all species have been assessed for the Red List more than once, reassessments are infrequent; birds are reassessed every 4–5 years with other groups reassessed less frequently. Therefore, some recent category changes may not yet be reflected or recorded as genuine changes. For example, the Iberian Lynx (*Lynx pardinus*) qualified for downlisting from Critically Endangered to Vulnerable on the Red List following a population increase from 94 to 2,021 individuals [77], but at the time of our analysis had not yet been reassessed and documented as a genuine change. Data on the drivers of improvements in status are currently available for even fewer species (birds and some mammals), making it impossible to distinguish the relative importance of multiple drivers of improvement where more than one factor is relevant. Going forward, as more species receive Green Status assessments [28,29], this will help overcome these shortfalls and enable the calculation of a key indicator to track species' recovery and to understand conservation impact, particularly compared with a pre-human baseline.

## Conclusions

Despite inevitable data gaps and uncertainties, we demonstrate the clear value of the Red List for understanding conservation action impact, including through documentation of the actions underway for species and genuine changes in species' status. Our results suggest that a wide range of conservation actions have successfully reduced species' extinction risk, particularly when targeted at specific species and locations, and have prevented extinctions of species at greatest risk. However, given relatively few species have shown signs of full recovery,

achieving Goal A of the Global Biodiversity Framework [4], which calls for restoration of species' populations to resilient levels as well as reduction of extinctions and extinction risk, will require substantially more ambitious, coordinated scaling-up of conservation interventions, particularly landscape-scale actions that benefit widely distributed species.

## Supporting information

**S1 Text. Classification of species groups.**
(DOCX)

**S2 Text. Classification of threats to species.**
(DOCX)

**S3 Text. Classification of actions in place for species.**
(DOCX)

**S4 Text. Breakdown of number species improving in status across species groups and number attributed to conservation or other means.**
(DOCX)

**S1 Table. Chi-squared statistics showing for each taxonomic group and IUCN Red List category, and IUCN Red List category within each taxon, whether different groups were more or less likely to have conservation actions in place.** For analysis of actions across IUCN Red List categories within taxa, this could not be evaluated for Merostomata, Myxini, Sarcopterygii and warm-water reef building corals.
(DOCX)

**S2 Table. Chi-squared statistics showing for all taxa, each taxonomic group individually, and by IUCN Red List category whether groups are more or less likely to have each type of conservation actions in place.** If actions are absent, they were not reported for any species in that group.
(DOCX)

**S3 Table. Cumulative linked mixed model output, showing how (i) animal species' global population trends, (ii) genuine changes in species' IUCN Red List category and (iii) prevented decline in species' state (estimated, Green Status of Species) varies with different species' traits, threats and actions.** Positive estimates indicate the variable correlates with a more favorable state, with positively and negatively correlated variables in green and orange font, respectively. The below shows overall model results, models just looking at only species traits, and models for particular taxa with sufficient species and actions to evaluate independently.
(DOCX)

**S4 Table. Sensitivity analysis for species conservation state model selection based on both forward and backwards AICc selection for models of species' global population trend, genuine change in Red List category and prevented decline in species' state (estimated, Green Status of Species).**
(DOCX)

**S5 Table. Chi-squared post-hoc tests of actions which led to improvements in bird species' IUCN Red List category compared to the other actions.** Mammals had no significant differences between actions.
(DOCX)

**S1 Fig. The proportion of species with reported conservation actions in place across different animal taxonomic groups that have been comprehensively assessed on the IUCN**

**Red List, split by their IUCN Red List category.** The dashed red line indicates the mean proportion across all species. See FigS1.csv for underlying data, available at: https://www.iucnredlist.org/resources/data-repository#Past%20conservation%20efforts.
(TIF)

**S2 Fig.** The percentage of species in animal groups which have been comprehensively assessed by the IUCN Red List that have each type of conservation action in place, (**A**) split by taxa that have been comprehensively assessed in the IUCN Red List (more than 80% assessed, excluding Cephalopoda, Merostomata, Myxini, Petromyzontid and Sarcopterygii) and (**B**) split by IUCN Red List category. See Fig S2A.csv and Fig S2B.csv respectively for underlying data, available at: https://www.iucnredlist.org/resources/data-repository#Past%20conservation%20efforts.
(TIF)

## Acknowledgments

Thanks to members of the Conservation Science lab group in the Zoology Department in Cambridge, A. Manica and E. Turner for comments on preliminary results and M. Hoffmann for comments on a draft of the manuscript. Thanks to T. Worthington for help producing a fishnet for the species richness maps, and A. Christie and W. Morgan for modeling advice. Thanks also to all those who have contributed to the IUCN Red List of Threatened Species.

## Author contributions

**Conceptualization:** Ashley T. Simkins, William J. Sutherland, Lynn V. Dicks, Silviu O. Petrovan.

**Data curation:** Ashley T. Simkins, Craig Hilton-Taylor, Molly K. Grace.

**Formal analysis:** Ashley T. Simkins.

**Funding acquisition:** Ashley T. Simkins.

**Investigation:** Ashley T. Simkins.

**Methodology:** Ashley T. Simkins, William J. Sutherland, Lynn V. Dicks, Silviu O. Petrovan.

**Project administration:** Ashley T. Simkins.

**Resources:** Ashley T. Simkins, Craig Hilton-Taylor, Molly K. Grace.

**Software:** Ashley T. Simkins.

**Supervision:** William J. Sutherland, Lynn V. Dicks, Silviu O. Petrovan.

**Validation:** Ashley T. Simkins.

**Visualization:** Ashley T. Simkins.

**Writing – original draft:** Ashley T. Simkins.

**Writing – review & editing:** Ashley T. Simkins, William J. Sutherland, Lynn V. Dicks, Craig Hilton-Taylor, Molly K. Grace, Stuart H. M. Butchart, Rebecca A. Senior, Silviu O. Petrovan.

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
