## [Editor Report · Decision Letter 0]

21 Aug 2024

Dear Dr Simkins, 

Thank you for submitting your manuscript entitled "What works to improve species conservation state? An analysis of species whose state has improved and the actions responsible" for consideration as a Research Article by PLOS Biology.

Your manuscript has now been evaluated by the PLOS Biology editorial staff, as well as by an academic editor with relevant expertise, and I'm writing to let you know that we would like to send your submission out for external peer review.

IMPORTANT: We think that your paper would be best considered as a Short Report. Please select "Short Reports" as the article type when you upload your additional metadata (see next paragraph).

Once your full submission is complete, your paper will undergo a series of checks in preparation for peer review. After your manuscript has passed the checks it will be sent out for review. To provide the metadata for your submission, please Login to Editorial Manager (https://www.editorialmanager.com/pbiology) within two working days, i.e. by Aug 23 2024 11:59PM.

Kind regards,

Roli Roberts

Roland Roberts, PhD

Senior Editor

PLOS Biology

rroberts@plos.org

---

## [Decision Letter · Decision Letter 1]

29 Oct 2024

Dear Dr Simkins,

Thank you for your patience while your manuscript "What works to improve species conservation state? An analysis of species whose state has improved and the actions responsible" went through peer-review at PLOS Biology. Your manuscript has now been evaluated by the PLOS Biology editors, an Academic Editor with relevant expertise, and by two independent reviewers.

You'll see that reviewer #1 is very positive, but has a number of concerns; most of these can be addressed textually, but his/her prime concern is your decision not to consider ex situ conservation measures. Reviewer #2 is also positive, and has no issues with the models and methodology, but thinks that the manuscript would benefit from shortening considerably to enhance its appeal (e.g. halve the length of the Results, and remove repetition from the Discussion). S/he also wants you to discuss the limitations of using the Red List for this analysis, and to emphasise the distinction between populations and species. A couple of the points might require minor extra analyses.

I discussed the reviews with the Academic Editor, who said "As the authors can see, both referees are enthusiastic about the manuscript, but recommend some revisions. Referee 2 especially brings up specific, but also general points about manuscript length, disclaimers about the IUCN dataset, and most importantly, the third point about populations versus species. Overall, I suggest the authors address referee comments and resubmit."

In light of the reviews, which you will find at the end of this email, we are pleased to offer you the opportunity to address the comments from the reviewers in a revision that we anticipate should not take you very long. We will then assess your revised manuscript and your response to the reviewers' comments with our Academic Editor aiming to avoid further rounds of peer-review, although might need to consult with the reviewers, depending on the nature of the revisions.

**IMPORTANT - SUBMITTING YOUR REVISION**

*Resubmission Checklist*

*Published Peer Review*

*PLOS Data Policy*

*Blot and Gel Data Policy*

Sincerely,

Roli Roberts

Roland Roberts, PhD

Senior Editor

PLOS Biology

rroberts@plos.org

REVIEWERS' COMMENTS:

Reviewer #1:

It was a pleasure to read and review this manuscript. This is the first time I have ever written that but it's true. The paper is exceptionally well written. The language is clear, transparent and unambiguous. The presentation of the analysis and results was straightforward and logical. My congratulations and appreciation to the authors.

Below are a few comments for consideration:

* It is great that the authors are mining the IUCN Red List for useful information that can help us do more and better conservation. This is, of course, what the database is for. However, terms and definitions are important here, not least because they get very careful treatment in the Red List database, which is one of the things that makes it so valuable. In the abstract, the authors state, "This suggests a range of conservation interventions have successfully conserved some species at greatest risk but have rarely recovered populations to resilient levels." This seems contradictory. Is it saying you can successfully conserve species without recovering them to resilient levels? 

* Invasive Species Control seems like an outcome to me, not an action, as does "potential occurrence in protected areas". What does "potential occurrence" mean, as an action? From the manuscript's title, and the introduction, I was expecting an analysis of more specific actions that could (for example) give us new insights into how to achieve those outcomes. I think what this analysis is really telling us is that when we successfully mitigate threats and take positive action (like reintroduction), species bounce back, but this work is easier, and the bounce back is quicker, when we only have to do it over relatively small, bounded area for a specific population. This is interesting but not novel. 

* My main critique is that the authors erred when choosing to exclude ex situ conservation from their analysis: "Ex situ conservation (captive breeding) was excluded from the models as it does not have an impact on species' extinction risk until individuals are introduced into the wild (which is documented separately as 'species reintroduction' or 'benign introduction')" (my emphasis). It would be preferable if, wherever it says, 'extinction risk', it said instead, 'risk of extinction in the wild', or something similar. The Red List distinguishes between Extinct, and Extinct in the Wild, and the short cut is problematic as it suggests ex situ makes no difference to extinction risk. Some species cannot be reintroduced for decades, so it's not a trivial distinction.

o The analysis showed that "Reintroductions or translocations were significantly more likely to have led to reductions in extinction risk compared with other actions",

o We can probably assume that many reintroductions were from ex situ sources (though they didn't measure this),

o So, a threatened species with a healthy ex situ population will have more reintroduction options than one without,

o Therefore, that species must have less risk of extinction than if there were no ex situ population in place.

The wording they use probably just reflects RL terminology. Strictly speaking, the RL measures risk of becoming Extinct in the Wild, rather than the risk of extinction (usually one and the same thing, but not always). But the authors express it in a more generalized way: "does not have an impact on species' extinction risk".

* This paper makes clear the need for the Red List Conservation Actions work currently underway to result in a practical, comprehensible set of standards.

* Given the fact that a shockingly low 51.8% of species in comprehensively assessed animal groups have conservation actions in place recorded, I would like to see the authors call for a change in instructions to RL assessors making recording of conservation actions in place required rather than recommended.

Reviewer #2:

I have reviewed the manuscript "What works to improve species conservation state? An analysis of species whose state has improved and the actions responsible". Using IUCN Red List data for almost 70,000 species, the manuscript aims to link the conservation status of species with the conservation actions implemented and threats affecting those species. Overall, this is a very interesting and relevant study that provides really important insights into how our conservation responses relate to the state of biodiversity. Towards this, the manuscript provides a very large-scale perspective. For obvious reasons that approach cannot explore important details at the local level but potentially provides some overarching observation that are useful for local scale approaches. I think the topic and quality of the analysis makes this paper very fit for PLOS Biology, and I overall want to commend the authors on a great study.

The methodology seems appropriate and I have no concerns about how the study has been conducted - which is probably also because the most interesting of the results are largely descriptive and hinges on how the Red List data is categorized (which follows a mixed of using established categories and logical decisions to group within those) and grouped. The models also seem appropriate given the data. 

However, I have three overall concerns as well as some minor comment from reading the manuscript. None of the overall comments I consider to be particularly major - though still important to consider and ideally address.

First, the manuscript is very long. The Discussion for example is just over 10 pages and the Result section (without figures) is ca. seven pages. The Introduction seems appropriate and is a good setup for the paper, but I would suggest that Methods, Results, and Discussion could be significantly reduced (maybe even by half) to provide a more focused and clearer narrative. This may also require removing some element, but if it helps provide a clearer story it might be worth it. I will leave the comment on length at that. I do not believe it is my role as a reviewer to decide the focus on someone else's paper, that should be the prerogative of the authors, but I do believe the impact and uptake of this paper will be severely reduced if the main story cannot be presented in a shorted and more precise manor. And given that I really like the work that has been done, I would think that a shame. Related to this - while I agree that a key role of the Discussion is to highlight the main results of the paper and put them in context of existing knowledge, I did find the Discussion a bit heavy on repeating things already presented in the Results section. I would suggest this is reduced and that the Discussion is reserved to discuss the most interesting results not necessarily all results and to bring them out in a way that highlights their importance rather than repeating numbers already presented in the previous section.

My two other comments related to things I believe needs to be explained and discussed a bit more - and yes I realize that asking in one comment to cut and in the next to add might not make the task of the authors any easier.

Second, I believe the Red List is an appropriate data-source for this analysis, and importantly the only data source that can be used to answer these questions at this scale. However, that does not mean it is perfect. I think the Discussion needs to include a paragraph of self-reflection on the weaknesses of the Red List for this analysis and the implications of this for the interpretation of the study.

Third, except for maybe the most threatened (and small ranging species) conservation actions are implemented at the level of populations not species. Thus, the paper is making a leap in connecting species and actions. Given the data, this leap is probably unavoidable, but it would be good to reflect on this in the Discussion. There is maybe a bit of this in the part specifically about protected areas and how an overlap in range might not equate to an overlap in focus, but this is a more overall issue for the manuscript.

Minor comment:

Line 42: technically, its not the world but the signatories or nations of the Convention on Biological Diversity that committed to this - I realize there is a big overlap, but still using CBD language here would be more appropriate.

Line 78: "IUCN" missing before "Red List" - check throughout the manuscript.

Line 78-82: the main message of this sentence is not entirely clear to me. The Red List should (hopefully) document which species have undergone *genuine* changes in their extinction risk without linking it to actions. I am sure there is something interesting here to include, but in its current form, this sentence seems to mix things without clearly articulating their connection.

Line 86: check presentation of IUCN reference (and in text-references in a few places seems to have different formats - for example sometimes including the initials of first and middle names).

Lines 88-90: not entirely clear to me. Is this a general criticism of the Red List? In that case a bit more nuance might be needed.

Lines 112-116: should numbers be given with an "n" - i.e. "(n = 7,983)"?

Line 125: rewrite - maybe: "For just under half of the mammals, the IUCN Red List did not contain information about generation length, so …."

Line 129: I would suggest using parenthesis around scientific name for the Greenland shark. Also relevant other places in the manuscript.

Lines 126-129: could probably be deleted - you have described above that you checked the data with the IUCN Red List unit, I guess this is just one particular case of what would be reasons for checking.

Lines 134-135: which equal area projection?

Lines 137-140: this assumption is not without its limitation. I do not suggest changing anything analytically, but maybe including an acknowledgement, in the Discussion section, that there is a leap here. 

Lines 290-292: I realize that different parameters have been presented in other parts of the methods in sections detailing how they were developed. But it would be useful to recap here what is the total list of variables included in the full model.

Lines 292-294: I am not a huge fan of conducting only backwards model selection. I would suggest at the very least that both backwards and forward model selection was performed to ensure that they reach the same model, or even better that a method like dredging or Lasso is used (Lasso might be most appropriate especially if the authors are concerned about the "fishing expedition" risk of dredging.

Lines 344-346: I was a bit confused by the introduction of "other invertebrates" here as the first part of the sentence reef-building corals - which I guess technically are invertebrates but which are rarely referred to as such. Maybe drop the "other"?

Lines 485-287: this paper present some really interesting results. The first sentence of the discussion should really re-state what the main take-home message is. However, by relating the finding to how they are only marginally different to another study (i.e. Senior et al. 2024) that effect is lost. I would strongly suggest rephrasing to relate to the findings of this study in their own right.

Line 500: each species had more actions or across the species there were more actions? This needs to be clarified. 

Line 517: delete the first "groups"

Lines 526-528: good point and one I would relate to the over-representation of LC species in protected areas, as this is likely to be more the case for LC species and threatened species and thus, the importance of PAs is likely more inflated for that group.

Line 529: these numbers need to be updated. Especially the marine coverage statistics is wildly outdated.

Figure 1: check the numbers close to the x-axis and fix them where they are not fully visible.

Figure 2: I think these are interesting and probably good as main figures, but richness map of any sub-division below all species is always very likely to be strongly correlated to species richness. I can see strong deviations here from that pattern, but I would suggest presenting in the supplementary material a map of the residuals between all species richness and changes in Red list status to show highlight where chose changes really differ from the predictions based on SR.

Figure 3: I really like the idea of this figure, but also have to admit I didn't find it super easy to decode. Starting with the three categories on top that are not super easy to place in relations to the columns below. I also hadn't from the method seen that Red List category was included in five different versions. What is the rationale for this? what theory supports trying red list category to the power of 5? Same with Initial Red List category? This seems overly complicated.

---

## [Decision Letter · Decision Letter 2]

18 Jan 2025

Dear Dr Simkins,

Thank you for your patience while we considered your revised manuscript "What works to improve species’ conservation status? An analysis of species that have improved in status and the actions responsible" for publication as a Short Reports at PLOS Biology. This revised version of your manuscript has been evaluated by the PLOS Biology editors and one of the original reviewers.

Based on the reviews, we are likely to accept this manuscript for publication, provided you satisfactorily address the remaining points raised by reviewer #2. I should note that we were unable to contact the Academic Editor on this occasion, so it's possible that they will send me some further requests that I shall forward to you. Please also make sure to address the following data and other policy-related requests.

IMPORTANT - please attend to the following:

a) Please can you change your Title to something with an active verb and no punctuation? We suggest something like "Past conservation efforts reveal which actions lead to positive outcomes for endangered species"

b) Please address the remaining concerns from reviewer #2. Regarding their continuing request for simplification of the Discussion, while we don't have a formal word limit for this article type, you should aim to maximise the accessibility and appeal for our readership; removing repetition sounds like a sensible move at least.

c) Please address my Data Policy requests below; specifically, we need you to supply the numerical values underlying Figs 1ABC (this might actually be in the Fig), 2AB, 3, 4, S1, S2AB, either as a supplementary data file or as a permanent DOI’d deposition.

d) Please cite the location of the data clearly in all relevant main and supplementary Figure legends, e.g. “The data underlying this Figure can be found in S1 Data” or “The data underlying this Figure can be found in https://zenodo.org/records/XXXXXXXX

e) Please make any custom code available, either as a supplementary file or as part of your data deposition.

We expect to receive your revised manuscript within two weeks. 

*Published Peer Review History*

*Press*

Sincerely,

Roli Roberts

Roland Roberts, PhD

Senior Editor

rroberts@plos.org

PLOS Biology

DATA POLICY:

Regardless of the method selected, please ensure that you provide the individual numerical values that underlie the summary data displayed in the following figure panels as they are essential for readers to assess your analysis and to reproduce it: Figs 1ABC (this might actually be in the Fig), 2AB, 3, 4, S1, S2AB. NOTE: the numerical data provided should include all replicates AND the way in which the plotted mean and errors were derived (it should not present only the mean/average values).

CODE POLICY

DATA NOT SHOWN?

REVIEWER'S COMMENTS:

Reviewer #2:

I have now been through the updated version of "What works to improve species' conservation status? An analysis of species that have improved in status and the actions responsible" (PBIOLOGY-D-24-02262R2). 

I overall think the authors have done a good job of responding to the reviewers' comments and I think the manuscript is improved because of this. I don't have any major issues with the updated manuscript and I think the quality and clarity as well as the scope now make this a great paper for publication in PLOS Biology. And I again want to congratulate the authors on this work and I am looking forward to seeing it out, and I would strongly recommend this work be published in PLOS Biology.

My only overarching comment is very similar to my comment on the original manuscript and relate to length. Even though I can see the authors have done a lot to address my previous comment on this (in particular in relation to the Result section), the manuscript still feels quite long with the Discussion section still close to 10 pages. I will leave this between the editor and authors. To me the discussion still, in places a bit too repetitive of the results. As a consequence, the reader is not helped very well in terms of clearly identifying and navigating what the authors see as the main and most important take-home messages. But I can also see that there is many interesting points being made, so if the authors feel this is the right format for their paper, then I will not argue with that - even if I would likely have preferred a version that was still significantly shortened compared to even the new version.

A few minor observation when reading it:

Line 157: helena in " Troides Helena" should not be capitalized. 

Lines 269-275: would it make more sense to just include references to software in the sections where their use is described rather than having six lines dedicated to this.

Line 303: are the numbers within parenthesis the number of species? Then I would include "n=" so it reads: (n = 1,220) - for example.

Lines 321-328: this seems very interesting but also currently quite confusing. For example, how are these two statements "contrasting"?: 1) "Species with smaller global ranges, those at lower risk of extinction and those found in both terrestrial and marine ecosystems were more likely to have increasing global populations." And 2) "species with larger global ranges, those previously at higher risk of extinction and marine species were more likely to have improved in Red List category". Both seems to be encouraging even if the specific metric of improvement is different (i.e. population increases vs. improved red list category).

Lines 374-378: can this sentence be broken up into multiple?

Line 380: I think "are" should be deleted? Or "with" replaced with "where"?

Lines 434-444: still seems quite repetitive of the result section. Is there no way the parts that repeats the results could be even shorter?

Line 493: how does a new section start with "Similarly"?

---

## [Editor Report · Decision Letter 3]

3 Feb 2025

Dear Dr Simkins,

Thank you for the submission of your revised Short Report "Past conservation efforts reveal which actions lead to positive outcomes for species" for publication in PLOS Biology. On behalf of my colleagues and the Academic Editor, Uma Ramakrishnan, I'm pleased to say that we can in principle accept your manuscript for publication, provided you address any remaining formatting and reporting issues. These will be detailed in an email you should receive within 2-3 business days from our colleagues in the journal operations team; no action is required from you until then. Please note that we will not be able to formally accept your manuscript and schedule it for publication until you have completed any requested changes.

IMPORTANT: I've asked my colleagues to include the following request with their own: "I note that your data/code provision for this study will be deposited in the IUCN Red List Data Repository. I will need to assess provision using the final URL before I can approve publication, so please provide this ASAP. Also please include this link in *all* relevant Figure legends (those for Figs 1ABC, 2AB, 4, S1, S2AB); I know this is repetitive, but it makes the Figs more stand-alone."

Sincerely, 

Roli Roberts

Senior Editor

PLOS Biology

rroberts@plos.org